# Association between White Matter Hyperintensities Burden and Cognitive Function in Adult Asymptomatic Moyamoya Disease

**DOI:** 10.3390/jcm12031143

**Published:** 2023-02-01

**Authors:** Jun Shen, Ziwei Xu, Zhengxin Liu, Yu Duan, Wenshi Wei, Jie Chang

**Affiliations:** 1Department of Neurology, Huadong Hospital of Fudan University, Shanghai 200040, China; 2Department of Neurosurgery, Huadong Hospital of Fudan University, Shanghai 200040, China

**Keywords:** asymptomatic Moyamoya disease, cognitive impairment, white matter hyperintensities, executive function, semantic memory

## Abstract

***Background and Purpose:*** White matter hyperintensities (WMH) caused by chronic cerebral hypoperfusion are common in Moyamoya disease (MMD) patients, but WMH burden with comprehensive cognition in adult asymptomatic MMD remains unknown. This study tried to investigate the association between the WMH burden and cognitive function in adult asymptomatic MMD. ***Methods:*** Sixty-four adult asymptomatic MMD patients without surgical revascularization were enrolled in this study and underwent a 3T MRI scan and complete cognitive tests from 2021 to 2022. WMH volume was extracted with brain anatomical analysis using the diffeomorphic deformation (BAAD) toolbox, which works on SPM 12 software. Multivariable linear regression analysis was performed to assess the association between WMH burden and cognitive function in asymptomatic MMD. ***Results:*** Firstly, our data showed that lower education levels and higher WMH burden were strongly related to global cognitive impairment after adjusting for other variables. Secondly, WMH severity was significantly associated with several domains of neurocognitive function, including memory, semantic memory, and executive function. Finally, when stratified by sex, the female participants with WMH severity had lower cognitive performance in all areas than male participants. ***Conclusions:*** These results suggest that WMH burden was highly correlated with global cognition, memory, semantic memory, and executive function in asymptomatic MMD. Especially in female participants, the relationship became more evident.

## 1. Introduction

Moyamoya disease (MMD) is a rare cerebrovascular disease with progressive stenosis at the terminal portion of the internal carotid artery and the compensatory proliferation of small puffy vessels in the base of the brain [1]. These small puffy vessels are prone to infarction or rupture, which can lead to an ischemic/hemorrhage stroke [2]. Recently, the cognitive deficits in ischemic MMD and hemorrhage MMD have been well established [3,4,5]. However, asymptomatic MMD usually occurs more insidiously than ischemic/hemorrhage MMD, whereas chronic cerebral hypoperfusion in asymptomatic MMD might also precipitate cognitive impairment [6]. Until now, limited attention has been paid to the profile of cognition in asymptomatic MMD.

White matter hyperintensities (WMH) are frequently observed on the cerebral MRI of elderly individuals [7]. The reduction of cerebral perfusion may induce WMH, which is associated with decreased global cognitive performance, executive function, and memory [8]. There is converging evidence about the relationship between white matter impairment and cognitive impairment in ischemic MMD [3,9]. However, no study has definitely investigated the association between WMH burden and cognitive function in adult asymptomatic MMD. 

This study first tried to evaluate the cognitive profile of adult asymptomatic MMD and then determine the association between WMH burden and cognitive function in adult asymptomatic MMD.

## 2. Materials and Methods

### 2.1. Participants

Participants diagnosed with asymptomatic MMD (Suzuki grades 3) at the Neurosurgery Department of Huadong hospital were enrolled in this study from January 2021 to August 2022. Written informed consent was obtained from the participants. The study was approved and reviewed by the ethics committee of Huadong Hospital of Fudan University (No. 2018030). The inclusion criteria were as follows: (1) Patients should meet the diagnosis guidelines of MMD based on digital subtraction angiography without infarcts and hemorrhage. (2) Patients were willing to take neuropsychological tests and sign the written informed consent. (3) No neurologic deficits were found in the MMD patients by physical examination. The exclusion criteria were as follows: (1) Ischemic MMD, hemorrhage MMD, and moyamoya syndrome were excluded. (2) The educational backgrounds of patients were excluded below middle school. (3) Patients should have no additional neurological disease history, such as brain tumors, neurodegenerative diseases, demyelinating diseases, and seizures. (4) Patients with psychiatric diseases were excluded. 

### 2.2. Demographic and Vascular Risk Factor Information

The demographic information was obtained from the participants, including age, sex, and educational background. Specific vascular risk factors information, including smoking, hypertension, diabetes mellitus, and lipidaemia, were recorded. 

### 2.3. Neuropsychological Assessment

All participants were evaluated with Montreal Cognitive Assessment (MoCA) [10], Hopkins Verbal Memory Tests-Revised (HVLT-R) [11], the Verbal Fluency Test (VFT) [12], Trail-Making Test-A (TMT-A), Trail-Making Test-B (TMT-B) [13].

MoCA is widely used to assess global cognition. The former HVLT is conducted to appraise recognition memory immediately after learning trials, while the latter (HVLT-R) appraises delayed recognition. The VFT reflects the integrity of semantic memory. The visuomotor processing speed is related to the TMT-A score, while an executive function is associated with the TMT-B score. 

### 2.4. MRI Acquisition

All MR images were acquired on 3.0 Tesla MRI (MAGNETOM Trio, Siemens, Erlangen, Germany) in the Department of Radiology of Huadong Hospital, Shanghai. T1-weighted images, T2-weighed images, and T2 fluid-attenuated inversion recovery (FLAIR) sequences were included in this study. The T2 FLAIR sequence was used to measure WMH with the following scan parameters: TR/TE = 8500/81 ms; inversion time = 2000 ms; flip angle = 150°; slice thickness = 5 mm. 

### 2.5. WMH Volume

The methods used to quantitatively measure WMH volume and the total intracranial volume based on T1 and T2 FLAIR images have been described previously [14]. Firstly, the Digital Imaging and Communication on Medicine software (DICOM) of T1 and T2 FLAIR images were transformed to the Neuroimaging Informatics Technology Initiative (NIfTI) for preprocessing. Secondly, the brain was segmented to obtain the gray matter, white matter, and cerebrospinal fluid on T1 images. Finally, WMHs were checked and segmented on T1 and T2 FLAIR images according to STRIVE guidelines (Standards for Reporting Vascular Changes on Neuroimaging) [15]. 

### 2.6. Statistical Analysis

All analyses were conducted using the SAS program (version 9.4; SAS Institute, Carry, NC) unless otherwise specified; all tests were 2-tailed with α = 0.05. Continuous data and categorical data were expressed as mean (SD) and numbers (percentages) in the description of the baseline clinical characteristics of the patients. We applied multivariable linear regressions to estimate the association of white matter hyperintensities with cognitive function indices. We entered all the covariates into the model to make a mutual adjustment. The covariates included age, sex, education, hypertension, diabetes mellitus, hypercholesterolemia, smoking, and alcohol.

## 3. Results

The clinical characteristics of asymptomatic MMD participants (age, 45.8 years, SD 10.6; 43.8% female; 57.8% higher educational level) are listed in Table 1. The representative WMH is listed in Figure 1. The total MoCA score is 30 points. In this study, the mean MoCA score is 22.5. The mean HVLT-R total recall is 15.8 points, SD 8.3. The mean HVLT-R delayed recall is 10.5 points, SD 5.8. The mean VFT is 40.0 points, SD 9.7. The mean TMT-A is 69.8 s, SD 42.4. The mean TMT-B is 96.3 s, SD 62.5.

### 3.1. Associations between Risk Factors and Global Cognition in Asymptomatic MMD

Multivariable linear regression analysis showed that education (*p* = 0.0001) and WMH volume (*p* = 0.0006) were highly correlated with global cognition after controlling all other variables in Table 2, whereas other demographic factors and vascular risk factors were not associated with global cognition in Table 2.

### 3.2. Associations between WMH and Cognitive Subdomains in Asymptomatic MMD

Although negative correlations were found between WMH and global cognition, we tried to investigate the relationships between WMH and cognitive subdomains. In the multivariable regression model, after controlling demographic factors and vascular risk factors, WMH were significantly associated with memory (*p* = 0.0048), semantic memory (*p* = 0.0005), and executive function (*p* < 0.0001) in Table 3.

### 3.3. Interactions between Sex and WMH in Relation to Global Cognition in Asymptomatic MMD

This study enrolled 36 male participants and 28 female participants. When we stratified the participants by sex, we found that associations between WMH and global cognition became more obvious in the female participants than in male participants after controlling for all other variables of demographic factors and vascular risk factors in Table 4. 

## 4. Discussion

Our study investigated the relationship between WMH burden and cognition in asymptomatic MMD participants. We yield three important findings: firstly, we showed that education and WMH burden were associated with cognition even after controlling all other variables; secondly, we demonstrated that the WMH burden was related to memory, semantic memory, and executive function after controlling for all other variables. Finally, we found that WMH burden plays a vital role in the cognition of female participants compared to male participants.

This study shows that only a higher educational level is positively correlated with global cognition. This is consistent with previous research, which suggested that the relationship between educational attainment and cognitive function is established in all studied populations [16]. However, this study did not find relationships between other demographic factors, vascular risk factors, and global cognition. The reason could be due to the smaller sample in this study. As we can see, there is a trend that hypertension negatively correlates with global cognition (*p* = 0.06). 

This study suggests that WMH is an important risk factor for global cognition beyond the effect of individual vascular risk factors. White matter vascular changes, which induce cognitive impairment, are most prominent in vascular dementia [17]. The pathological change in WMH is the ischemic demyelination that disrupts the cortex-subcortical nuclei connections, which is correlated to dementia-related pathological processes [18]. Several lines of evidence found that WMH was associated with reductions in working memory, processing speed, and executive function in clinical studies [19,20,21]. 

Our study also comprehensively investigates the associations between WMH and cognitive subdomains in asymptomatic MMD. Mounting evidence has observed the cognition in MMD. Liu et al. investigated white matter impairment and cognitive dysfunction relationships in ischemic MMD patients [3]. Ken et al. found that revascularization surgery improves cognitive function, increases the fractional anisotropy of white matter, and increases brain connectivity in adult MMD [22]. Until now, fewer studies have focused on the relationships between WMH burden and cognition in asymptomatic MMD. Our study demonstrated that WMH burden correlated to memory impairment, semantic memory impairment, and executive dysfunction in asymptomatic MMD, which is in line with previous studies.

Furthermore, we report gender differences in the association between WMH and cognition. Generalized linear regression models showed that higher WMHs was significantly correlated with global cognitive impairment in females compared to male. These results were not consistent with previous findings. Sachdev et al. found WMHs to be correlated with a reduction in processing speeds in men only [23]. A large cross-sectional study suggested no association between WMH and cognition; however, white matter damage was associated with executive dysfunction only in men [24]. In cognitively unimpaired elderly, a higher WMH volume was found in women compared to men [25]. The reason for these conflicting results may be due to cognitive reserve. 

The results of our research should be interpreted with caution in light of its limitations. Firstly, our sample size is bigger than the previous study of MMD and cognition [26,27], but the participants were from one single medical center. There may exist selection bias. Secondly, it is a cross-sectional study with limitations in causal inference; longitudinal research could efficiently explain the dynamic process between WMH and cognition. Furthermore, our study did not enroll a healthy control, and it would be much better to include a corresponding healthy control for this study.

## 5. Conclusions

Our study demonstrates that the WMH burden was highly correlated with global cognition, memory, semantic memory, and executive function in asymptomatic MMD. Meanwhile, we underline the importance of WMH and cognition in females compared with male patients. Our study provides clinical evidence for associations between WMH and cognition in asymptomatic MMD.

## Figures and Tables

**Figure 1 jcm-12-01143-f001:**
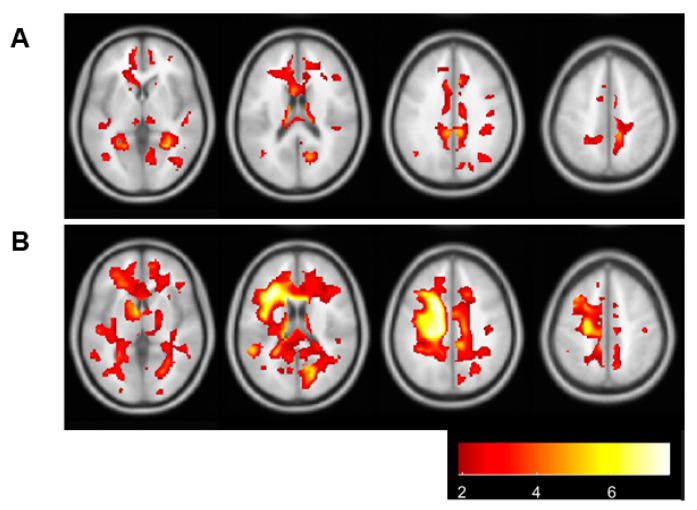
The results of WMH calculated by BAAD software. (**A**) Represents a patient with low WMH volume. (**B**) Represents a patient with high WMH volume.

**Table 1 jcm-12-01143-t001:** Demographic and clinical characteristics of participants.

	Mean (SD) or N (%)	n
**Demographic**		
Age	45.8 (10.6)	64
Sex, Female	28 (43.8%)	64
Education, High School, or College	37 (57.8%)	64
**Vascular risk factors**		
Smoking (Current)	17 (26.6%)	64
Alcohol (Current)	17 (26.6%)	64
Hypertension	28 (43.8%)	64
Diabetes Mellitus	7 (10.9%)	64
Hypercholesterolemia	14 (21.9%)	64
**MRI variables**		
WMH, cm^3^	6.6 (13.7)	57 *
**Cognitive tests**		
General Cognition (MoCA)	22.5 (5.1)	64
Learning (HVLT-R total recall)	15.8 (8.3)	64
Memory (HVLT-R delayed recall)	10.5 (5.8)	64
Semantic Memory (VFT)	40.0 (9.7)	64
Visuomotor Processing Speed (TMT-A)	69.8 (42.4)	64
Executive Function (TMT-B)	96.3 (62.5)	64

Abbreviations: WMH—white matter hyperintensity. * The other seven patients had an received MRI scan in the other hospital.

**Table 2 jcm-12-01143-t002:** Multivariable linear regressions showing associations between risk factors and global cognition.

Variables	β (95% CI)	*p* Value
Age	−0.05 (−0.16 to 0.05)	0.33
Sex	0.12 (−2.15 to 2.39)	0.92
Education	4.34 (2.11 to 6.57)	0.0001
Hypertension	−2.13 (−4.39 to 0.13)	0.06
Diabetes Mellitus	−1.59 (−5.40 to 2.21)	0.41
Hypercholesterolemia	−0.54 (−3.04 to 1.96)	0.67
Smoke (Current)	1.67 (−1.01 to 4.35)	0.22
Alcohol (Current)	0.58 (−1.99 to 3.15)	0.66
WMH	−0.15 (−0.23 to −0.06)	0.0006

Abbreviations: β indicates standardized beta coefficient; WMH—white matter hyperintensity.

**Table 3 jcm-12-01143-t003:** Associations between WMH and cognitive subdomains estimated by multivariable linear regression models.

Cognitive Variables	WMH	
	β (95% CI)	*p* Value
Learning (HVLT-R total recall)	−0.10 (−0.19 to −0.00)	0.04
Memory (HVLT-R delayed recall)	−0.14 (−2.23 to −0.04)	0.0048
Semantic Memory (VFT)	−0.28 (−0.43 to −0.12)	0.0005
Visuomotor Processing Speed (TMT-A)	0.32 (−0.46 to 1.09)	0.43
Executive Function (TMT-B)	2.00 (1.07 to 2.93)	<0.0001

Abbreviations: β indicates standardized beta coefficient; WMH—white matter hyperintensity; HVLT-R—Hopkins Verbal Memory Tests-Revised; VFT—Verbal Fluency Test; TMT-A—Trail-Making Test-A; TMT-B—Trail-Making Test-B. The analyses were adjusted for age, sex, education, hypertension, diabetes mellitus, hypercholesterolemia, smoke, and alcohol.

**Table 4 jcm-12-01143-t004:** Interactions between sex and WMH in relation to global cognition.

Sex	WMH	
	β (95% CI)	*p* Value
Male (n = 36)	−0.10 (−0.17 to −0.03)	0.0065
Female (n = 28)	−0.66 (−0.95 to −0.37)	<0.0001

Abbreviations: WMH—white matter hyperintensity. The analyses were adjusted for age, sex, education, hypertension, diabetes mellitus, hypercholesterolemia, smoke, and alcohol.

## Data Availability

The raw data supporting the conclusions of this article will be made available by the authors without undue reservation.

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
