# Peer review of "Association between White Matter Hyperintensities Burden and Cognitive Function in Adult Asymptomatic Moyamoya Disease"

_jcm, 2023, doi:10.3390/jcm12031143_

Round 1

Reviewer 1 Report

Study Summary:

The main purpose of this study was to review white matter hyperintensity (WMH) burden and cognitive function in adult patients with asymptomatic Moyamoya disease in Shanghai, China.  The WMH burden was calculated as a volumetric number based on the T2 and T1 weighted MRI scans.  A total of 64 patients were included in the study.  The authors performed multivariable linear regression and demonstrated that education level and WMH had a significant effect on global cognition.   They additionally find that several sub-domains of cognition including memory, semantic memory, and executive function differed based on WMH volume.  Lastly, they find that was more likely to play a role in female participants. 

Overall, this is an interesting study which makes practical sense.  They find that as one would expect education level impacts cognition which helps validate their model.  WMH has been found to be associated with global cognitive decline in conditions such as vascular dementia. This study finds that this association is also present in a small subset of asymptomatic MMD adult patients in a single hospital.  This is a useful article which makes clinical sense but is also careful not to over-state their findings.  

Critique:

I suggest that the authors include a figure showing how the WMH volume is calculated as this is what their entire study is based on. If they could, for instance show a figure that has a large volume and a small volume with an explanation of how these volumes were calculated it would help the readership have more confidence in their findings. Also, how were the tests administered? Was a blinded assessor used (for both the cognitive tests and the MRI interpretations)?  If not, then there is bias that could be present here and this would need to be clearly stated in a limitations section.

Author Response

Dear Reviewer,

Thank you for your valuable comments. 

Reviewer 2 Report

The authors present a study on WMH and cognition in asymptomatic MMD patients. The reason for the study is unclear and the effect that MMD plays cannot be determined since this study does not include healthy controls. Whether cognitive deficits or WMHs exists in MMD to a greater extent is not known based on this study, so the effect that MMD plays cannot be determined. In all the models, no metric of the disease is included despite the study being about MMD. There are several grammatical errors in the paper - it needs to be thoroughly revised. The discussion is rather limited. I have several comments outlined below: 

If these patients did not have stroke or ICH – how did they present to medical attention? Were these patients incidentally found to have MMD – if so, there is significant selection bias which should be stated in limitations. What were the reasons these patients presented to medical attention?

Patients were willing to take cerebral MRI scan in our hospital – this doesn’t really seem like an inclusion criteria

Patients should have no neurological diseases history such as brain tumor, neurodegenerative diseases, demyelinating diseases, and seizure – do the authors mean “no additional neurological diseases”

 Line 71 – all these tests should have references to support their use

 Line 90 – need a reference for STRIVE

 In table 1 – why did only 57 patients receive MRI, per the authors, the patients needed to have MRI as exclusion criteria

Line 99 – where did these covariates come from? Were they determined a priori?

 For cognitive tests in table 1, authors in caption or text should describe these texts – what is the total number of points for each test. For these tests, means are provided rather than medians, which might make better sense if the data are not normally distributed

 Line 102 – need a SD after reporting mean age

 In table 1, the authors state that % of HS and college education was 58% -- however, the authors in the methods excluded patients with below middle education. Does this mean nearly ~40% of patient had middle school only education? Did this play a role in informed consent processes?

 The authors make no mention of IRB information, required informed consent, etc in the methods

 Table 2- what is global cognition? Is this defined per MOCA scores? The authors do not make this clear – keep significant digits in p values the same in all tables 

This study is about MMD – but no MMD parameters are listed in tables; Suzuki grades etc; the purpose of studying WMH in MMD patients is a bit unclear. These results would seem to apply to all patients with any neurological disease. No healthy controls were used, so the effect of MMD cannot be determined. 

Discussion section , especially section on limitation is rather limited

Author Response

(The authors gave the same response as above.)

Round 2

Reviewer 1 Report

The authors have satisfactorily answered the comments.

Author Response

 Thank you very much for your valuable comments. 

Reviewer 2 Report

-       The authors did not correct the paper appropriately - For each of my original comments, the authors should include a revision in the paper and state where it was added. Only a cursory revision of the paper was completed. A summary of the changes below is listed, but please refer to my original comments. 

Authors did not include reasons for MRI in the paper –

-        Authors did not correct table 1 regarding MRI

-        Authors did not add explanation of how they decided on covariates for the model

-        Authors only added total score of MOCA – they should repeat for each test

-        IRB information not added in methods

-        The authors did not add Suzuki scores to the models

-        Paper is still rife with grammatical errors

Author Response

(The authors gave the same response as above.)
